# Impact of Dietary Habit, Iodine Supplementation and Smoking Habit on Urinary Iodine Concentration During Pregnancy in a Catalonia Population

**DOI:** 10.3390/nu12092656

**Published:** 2020-08-31

**Authors:** Maria-Teresa Torres, Lluis Vila, Josep-María Manresa, Roser Casamitjana, Gemma Prieto, Pere Toran, Gemma Falguera, Lidia Francés

**Affiliations:** 1Sexual and Reproductive Health Care (ASSIR), CAP Antoni Creus Querol, Catalan Health Institute, 08228 Terrassa, Spain; torrescostamt@gmail.com; 2Northern Metropolitan Research Support Unit (USR MN), University Institute in Primary Care Research Jordi Gol (IDIAP Jordi Gol), 08290 Cerdanyola del Vallès, Spain; ptoran.mn.ics@gencat.cat; 3Sexual and Reproductive Health Care Research Group (GRASSIR), University Institute in Primary Care Research Jordi Gol (IDIAP Jordi Gol), 08907 L’Hospitalet Llobregat, Spain; gfalguera.mn.ics@gencat.cat; 4Endocrinology and Nutrition Service, Sant Joan Despi Moises Broggi Hospital, 08970 Sant Joan Despi, Spain; Lluis.Vila@sanitatintegral.org; 5Departament of Nursing, Autonomous University of Barcelona, Bellaterra, 08193 Cerdanyola del Vallès, Spain; 6Multidisciplinary Research Group in Health and Society (GREMSAS), University Institute in Primary Care Research Jordi Gol (IDIAP Jordi Gol), 08007 Barcelona, Spain; 7Biochemistry and Molecular Genetics Service, Hospital Clinic, 08036 Barcelona, Spain; 8Health Assistance Management of Avila, 05001 Avila, Spain; gprietol@saludcastillayleon.es; 9Sexual and Reproductive Health Care (ASSIR), Northern Metropolitan Territorial Administration, Catalan Health Institute, 08202 Sabadell, Spain; 10School of Nursing, Faculty of Medicine and Health Sciences, Universidad de Barcelona, 08907 Hospitalet de Llobregat, Spain; lfrances@ub.edu

**Keywords:** pregnancy, iodine, dietary habits, iodine supplement, urinary iodine concentration

## Abstract

(1) Background: The nutritional status of women during pregnancy can have a considerable effect on maternal and fetal health, and on the perinatal outcome. Aim: to assess the changes occurring in dietary iodine intake, potassium iodide supplementation, and smoking habit, and the impact of these changes on the urinary iodine concentration (UIC) during pregnancy in a population of women in Catalonia (Spain). (2) Methods: Between 2009–2011, an observational study included a cohort of women whose pregnancies were monitored in the public health system in the Central and North Metropolitan areas of Catalonia. Women received individual educational counseling, a dietary questionnaire was completed, and a urine sample was collected for iodine determination at each trimester visit. (3) Results: 633 (67.9%) women answered the questionnaire at all 3 visits. The percentage of women with a desirable UIC (≥150 μg/L) increased from the first to the second trimester and remained stable in the third (57.3%, 68.9%, 68%; *p* < 0.001). Analysis of the relationship between UIC ≥ 150 μg/L and the women’s dietary habits showed that the percentage with UIC ≥ 150 μg/L increased with greater consumption of milk in the first trimester, and the same was true for iodized salt use in all three trimesters and iodine supplementation in all three. (4) Conclusion: During pregnancy, increased intake of milk, iodized salt, and iodine supplements were associated with an increase in the UIC.

## 1. Introduction

Iodine is an essential nutrient required for synthesizing thyroid hormones which regulate the body’s metabolism, growth, and development. Iodine requirements increase significantly during pregnancy and lactation, and if the intake is insufficient during these periods, thyroid hormone production can decrease, with consequent repercussions on the mother, the developing fetus, and the breastfed infant [1]. Severe or moderate iodine deficiency during pregnancy and lactation can affect the mother’s and newborn’s thyroid function, as well as the child’s later neuropsychological development [1]. According to the 2014 position statement of the Spanish Society of Endocrinology and Nutrition [2], several studies have shown that most Spanish mothers are iodine-deficient during pregnancy and lactation. Nonetheless, two recent studies performed in pregnant women in the first trimester, one in Asturias [3] and the other by our group in Catalonia [4], reported median urinary iodine levels of 197 μg/L and 172 μg/L, respectively, both values being within the margins the World Health Organization (WHO) considers indicative of adequate iodine nutrition (>150 μg/L). The foods that had the greatest impact on urinary iodine in these studies were iodized salt, milk, and also iodine supplements.

Appropriate nutritional habits together with iodine supplementation can help meet iodine requirements during pregnancy and postpartum, and prevent or correct iodine deficiency and its consequences [5]. Most foods contain relatively little iodine, and to ensure that the intake of this element is sufficient in each individual, the WHO and the United Nations International Children’s Emergency Fund (UNICEF) recommend universal salt iodization as a global strategy [6]. When iodized salt intake is of a voluntary nature, populations at higher risk, such as pregnant women, breastfeeding mothers, and children under two years of age, may not receive a sufficient amount of iodine unless periodic campaigns are conducted to encourage iodized salt consumption.

Health education regarding nutrition during pregnancy aims to improve the quality of the diet, instructing women on which foods and what amounts of food should be consumed to achieve an optimal dietary intake. It should also include advice on the use of various micronutrient supplements containing iron, folic acid, and iodine [7]. Regarding iodine, women who are planning pregnancy and those who are pregnant or breastfeeding should be encouraged to consume iodized salt, milk, and dairy products to obtain an adequate iodine supply. Potassium iodide (KI) supplements (150–200 µg/day) also guarantee the adequate intake of this element [2]. Another important aspect of health education during pregnancy is the recommendation to stop smoking. Among other harmful effects, tobacco is considered a goitrogenic substance that interferes with iodine uptake by the thyroid glands [8,9].

In Catalonia, the nutritional recommendations provided during pregnancy are aimed at maintaining a healthy weight, suppressing the use of tobacco and alcoholic beverages, and establishing a balanced, healthy diet. A healthy diet should be sufficient, complete, varied, and balanced, but also adapted to the characteristics of the individual, while being satisfying, safe, sustainable, and affordable. To cover iodine needs during pregnancy, the diet should contain 3 servings of milk and dairy products and 2 g of iodized salt per day [3,4,10]. If this iodine supply cannot be guaranteed, KI supplementation (200–250 μg/day) is recommended [10].

Few studies in our setting have focused on the changes in iodine nutrition that occur during pregnancy and the possible relationship between iodine status and dietary habits. Hence, this study aimed to investigate the effects of dietary iodine intake, KI supplementation, and smoking habit on iodine status in a population of pregnant women in Catalonia.

## 2. Methods

### 2.1. Design

An observational study carried out in a cohort of pregnant women monitored in the public health care system of the Central and North Metropolitan areas of Catalonia (Spain). Initially, the study was contemplated as part of a clinical trial project to evaluate the efficacy of an educational intervention imparted in groups to promote iodine intake and optimal urinary iodine concentrations (≥150 μg/L) in pregnant women. The women were enrolled, but the intervention by groups was found to be impracticable in our setting because of difficulties with ensuring the proper development of the educational program. Hence, the design was converted to an observational study with a single-arm (usual clinical practice) cohort in which women received the educational counseling on an individual basis. As a part of the routine clinical practice in our primary care centers, midwives impart information about diet and other relevant recommendations at a prenatal visit during the first trimester of pregnancy.

The study was conducted within the framework of the primary care center Program for Sexual and Reproductive Health Care (PASSIR) of the Catalonian Central and North Metropolitan regional offices of the *Institut Català de la Salut* (ICS, Catalan Institute of Health) in the province of Barcelona.

### 2.2. Participants

Consecutive recruitment was carried out during 2008 and 2009. All women older than 17 years attending the participating centers in their first trimester of pregnancy (<13 weeks) and willing to participate were included in the study. Pregnant women with thyroid disease, no telephone contact, or difficulty communicating with the health personnel (cognitive, sensory, or language problems), and those refusing to participate, were excluded.

### 2.3. Data Collection

Each trimester, a visit was scheduled in the participating primary care centers where women answered a questionnaire and a urine sample was collected for iodine determination. The questionnaire compiled sociodemographic data, including patient age, place of origin, place of residence (rural/urban), and educational level. Information on dietary habits and other factors was collected at a personal interview by primary care midwives, using a standardized questionnaire [11] that showed good reliability (the Cronbach’s alpha was 0.960, and the intraclass correlation coefficient was 0.927). The questionnaire contained items related to the consumption of cow milk (glasses/day; 1 glass = 200 mL), yogurt (units/week), cheese (portions/week), cooked vegetables, fresh vegetables, fish, canned fish, meat, cold cooked meats, eggs, fruit, and dried fruit (servings/week), regular consumption of iodized salt and daily use of iodine supplements (KI or iodine vitamin supplements) (Yes/No), and tobacco use.

Urinary iodine concentration (UIC) was determined as follows: A first-morning urine sample was collected from each woman, quickly frozen at −40 °C, and transported within 24 to 48 h to a central laboratory (Barcelona Hospital Clinic), where UIC determination (μg/L) was performed using the Benotti and Benotti method [12,13]. Urine was digested with chloric acid and then underwent the Sandell-Kolthoff reaction, in which iodine was determined by its action as a catalyst in the reduction of ceric ammonium sulfate in the presence of arsenious acid. The inter-assay and intra-assay coefficients of variation of the technique were 15.5% and 12.6%, respectively. Three times a year, the UIC assay undergoes evaluation by an external quality assessment program from the Spanish Association of Neonatal Screening (AECNE). UIC values obtained were dichotomized as <150 μg/L (insufficient) and ≥150 μg/L (adequate) [14].

### 2.4. Statistical Analysis

Quantitative variables are described as the mean and standard deviation or the median and first-third quartiles (Q1–Q3) for those with a non-normal distribution. Categorical variables are expressed as the absolute frequency and percentage. The Student *t*-test for independent data and the Mann-Whitney *U* test, as appropriate, were used to compare the quantitative variables, and the Pearson chi-square test was used for categorical variables.

The intake information was grouped into 3 categories (milk: 0, 1–2, >2 glasses; yoghourt: 0, 1–2, >2 units; cheese: 0, 1–2, >2 portions; and the remaining: 0, 1–2, >2 servings). Multilevel mixed-effects logistic regression models adjusted for paired measures in the 3 trimesters were performed to assess the effect of iodine-rich food intake or supplementation on UIC values ≥150 μg/L.

The initial regression model included all variables that were individually associated with the outcome at a significance level of *p* < 0.10. The final model included variables that were statistically significant at *p* ≤ 0.05; the Akaike information criterion and biological plausibility were also taken into consideration.

Missing values for food intake at follow-up were imputed using all available variables. The estimation model was based on 10 replicates and included the same variables as in the final hierarchical model.

Statistical significance was set at *p* ≤ 0.05 (two-tailed). Data were analyzed with SPSS (IBM Corp. Released 2017. IBM SPSS Statistics for Windows, Version 25.0. Armonk, NY, USA). Multiple imputation and mixed-effects logistic regression analyses were performed with the Stata/SE Statistical Package 14.0 (StataCorp LLC, College Station, TX, USA) for Windows.

### 2.5. Clinical Trial Registration

ClinicalTrials.gov: NCT01301768.

### 2.6. Ethical aspects

Ethics approval and consent to participate: All women were informed and signed their informed consent to participate in the project.

The study was approved by the Clinical Research Ethics Committee at the Primary Care Research Institute (IDIAP) Jordi Gol (Code: P07/02).

## 3. Results

In total, 970 women who attended a first-trimester prenatal visit were included. Among them, 945 (97.4%) answered the dietary questionnaire at this visit, 752 (77.5%) attended the second-trimester visit, and 705 (72.7%) the third-trimester visit (Table 1). The 633 (67.9%) women who answered the questionnaire at all 3 visits showed no significant socio-demographic differences nor differences for UIC relative to those who did not.

In the first trimester evaluation, the mean age was 30.6 (4.6) years, 83% of participants were Spanish natives, and 73.7% lived in an urban setting. As to educational level, 3.5% were illiterate or had not completed primary school, 24.6% had a primary school diploma, 41.6% a high school diploma, and 30.4% a university degree.

The results of the dietary questionnaire are summarized in Table 1. The column corresponding to the first trimester shows the dietary habits women had when they went to the first prenatal visit. At that time, they each received counseling on nutritional health according to the usual clinical practice. The columns for the second and third trimesters show the dietary habits acquired following educational counseling.

In general, the consumption of milk, cooked and fresh vegetables, fish, meat, eggs, and fruit increased along the three trimesters (*p* < 0.05) (Table 1). Yoghourt intake increased, but only to a marginally significant degree (*p* = 0.070). Consumption of cheese, cold cooked meats, and dried fruit showed no changes. Iodized salt use increased considerably in the second trimester and held steady in the third (35.7%, 85.4%, 88.9%; *p* < 0.001). The same trends were seen for iodine supplementation (46.8%, 88.8%, 87.6%; *p* < 0.001). Tobacco use decreased somewhat in the second trimester and was maintained in the third, with only marginal significance (24.1%, 20%, 19.5%; *p* = 0.056). The median UIC increased from the first to the second trimester and held steady in the third (172 μg/L, 210 μg/L, and 202.5 μg/L, respectively; *p* < 0.001).

### 3.1. Association with UIC ≥150 μg/L

Urinary iodine values were dichotomized into < 150 μg/L and ≥ 150 μg/L. The percentage of women with a desirable UIC level (≥150 μg/L) increased from the first to the second trimester and held steady in the third (57.3%, 68.9%, 68%; *p* < 0.001) (Table 1)

Analysis of the relationship between UIC ≥ 150 μg/L and the women’s dietary habits (Table 2) showed that the UIC increased as consumption of milk, fresh vegetables, and fruit increased in the first trimester, and the same was true for iodized salt consumption in all three trimesters and iodine supplementation in all three trimesters. Tobacco consumption was not related to the iodine status.

### 3.2. Multivariate Analysis

We carried out a multilevel mixed-effects logistic regression analysis of repeated measures to determine which intake-related factors were associated with UIC < 150 μg/L. Variables that were candidates for inclusion in the analysis were examined (Table 3). The resulting multivariate model (multilevel mixed-effects) is shown in Table 4A. A protective effect was manifested for iodine supplementation (OR = 0.42, 95% CI [0.34–0.52]; *p* < 0.001), iodized salt intake (OR = 0.67, 95%CI [0.54–0.83]; *p* < 0.001), and milk consumption (1–2 glasses: OR = 0.68, 95%CI [0.52–0.90]; *p* = 0.006; >2 glasses: OR= 0.64, 95%CI [0.45–0.90]; *p* = 0.011).

After carrying out multiple imputation of missing values in the three trimesters (Table 5) and repeating the process of variable selection, we obtained the same model as the previous one (Table 4B: multilevel with imputation), in which the protective roles of each of the factors were maintained. The model with imputation provided OR values and confidence intervals very similar to those of the model without imputation.

## 4. Discussion

The results of this study showed favorable changes in iodine nutrition in a sample of pregnant women from central Catalonia following individual educational counseling on dietary habits imparted by the midwife in the first trimester of pregnancy. The median UIC increased from the first to the second trimester and held steady thereafter (172, 210, and 202.5 μg/L, respectively). These changes were related to greater consumption of milk, iodized salt, and iodine supplements. The data indicate that iodine nutrition is adequate in pregnant women in our setting, in accordance with the recommendations of the WHO, the ICCIDD, and the UNICEF, which recommend a population median of > 150 μg/L [6].

The daily iodine intake recommended by the WHO during pregnancy and lactation is at least 250 μg/day. However, it is difficult to estimate iodine intake through analysis of the diet, as the amount of this micronutrient in both food and water can vary considerably from one area to another. Under normal conditions, there is a balance between iodine intake and urinary iodine elimination, which makes the determination of the UIC in casual urine specimens a reliable indicator of the iodine nutritional status of populations [15]. Evidently, a median UIC increase also implies a significant gain in the percentage of individuals with UIC values ≥ 150 μg/L. However, because of the considerable variability in urinary iodine level, this cut-off is not considered acceptable to differentiate between populations with adequate or poor iodine nutrition. Only the median UIC is deemed indicative of the iodine nutrition status in a specific population, such as pregnant women [16]. Nonetheless, some studies have shown that pregnant women with UIC < 150 μg/L can have a higher risk of goiter [17]. Prematurity and low birth weight are more prevalent in their newborns [18,19], and their children may show a lower IQ (median < 150 μg/g creatinine) [20,21] and greater oxidative stress [22]. Hence, it is likely that women with urinary iodine below this cut-off have a higher risk of iodine deficiency and greater morbidity, which can also extend to their offspring [23]. In the present study, the median UIC was adequate, but a substantial percentage of women (44.7% in the first trimester and 32% in the third) had levels < 150 μg/L and, likely, a greater risk of morbidity.

The probability of developing iodine deficiency-related disorders is undoubtedly higher when the median UIC does not reach 150 μg/L. However, a large part of the studies conducted in Europe over the last few years have reported median UIC values considerably lower than this limit, and even below 100 μg/L [24,25,26,27,28,29,30,31,32]. Only a few countries such as Iceland [33] and Holland [34] have reported median values in pregnant women higher than 150 μg/L. Thus, many of the studies conducted provide evidence that most pregnant women in Europe have insufficient iodine intake even in areas with adequate iodine nutrition [35]. Our results coexist with findings of excellent iodine nutrition in children in our setting [36] and adequate nutrition in the general population [37,38], despite the borderline situation in women of childbearing age.

The UIC changes observed over pregnancy have been a subject of debate, particularly in iodine-deficient areas. Some studies, such as one conducted in the United Kingdom [24], have shown a significant increase in the median UIC (from 42 μg/L in the first trimester to 69.4 μg/L in the third, or a creatinine increase from 103 μg/g to 126 μg/g). These changes were associated with an increase in the consumption of dairy products as the pregnancy progressed. Similar findings have been reported in studies from Norway (from 66 μg/L to 92 μg/L) [25] and Ghana [39], and in other studies carried out in Spain (Basque Country [40], Jaén [41] and Osuna [42]). In the three Spanish studies, the increase was clearly related to KI supplementation, received by most of the women included. Of note, in the study performed in Jaén, women who consumed iodized salt during the year before they became pregnant maintained a median UIC >150 μg/L over the entire pregnancy. In contrast, other studies, such as those by Stilwell [43], Brander [44], and Koukkou [45], have reported that UIC decreases with the progression of the pregnancy. This would be explained mainly by the depletion of iodine deposits due to increased requirements without adequate dietary compensation. In addition, the drop in UIC could be enhanced by pregnancy-related physiological changes in the glomerular filtration rate. Other studies, however, have reported no significant changes in the UIC over the three trimesters. [26,46].

In the present study, the dietary habits that had an impact on the UIC were consumption of milk and iodized salt, as well as KI supplementation. These findings concur with the results obtained in the study by Bath [24] in the United Kingdom, where daily milk intake (>280 mL) and consumption of Brazil nuts had an impact on the UIC. Among the pregnant women studied, 97% did not receive iodine supplements and only 6% used iodized salt. Similar findings were reported in the study by Dhal [25] in Norway, which showed that the dietary habits with the greatest effect on the UIC were consumption of dairy products and iodine supplementation (15.1% of pregnant women). There was no information on iodized salt consumption in that study. The authors found that the intake of milk and dairy products had a greater impact on the IUC than seawater fish and other marine products. In the study by Henjum [26], the factor with the greatest impact on the IUC during pregnancy was iodine supplementation, whereas in the study by Shashi Kant [47] in India, it was iodized salt intake (90.9% of pregnant women), although the median amount of iodized salt consumed was high, at 8.3 g/day.

Dairy products are an important source of iodine in numerous European countries, as evidenced in the study by Bath in the United Kingdom [24], Dahl in Norway [48], and some studies performed in Spain. Two Spanish studies have reported that milk is an important source of dietary iodine [49,50]. In our country, a glass of ordinary milk (200–250 mL) provides an average of 50 µg of iodine, or 20% of the recommended amount for pregnant women and breastfeeding mothers. The study by Menéndez [3] also showed an increase in the UIC with an intake of 2 or more daily servings of milk or milk products.

In the present study, vegetable consumption did not present in the final model as having a significant impact on the IUC, in accordance with the 2007–2012 study by NHANES showing no correlation between vegetable intake and the UIC [51]. The fortification of certain vegetables with iodine has been successfully undertaken to increase iodine intake [52], but this option is not contemplated in our setting.

Nutritional education and counseling during pregnancy are focused on improving the quality of the diet. Women are provided information on what foods and what amounts of food are needed to achieve proper nutrition, and they are instructed on the use of micronutrient supplements such as iodine when adequate amounts are not guaranteed by the diet [7]. We found an increase in the consumption of iodine-rich foods from the first to the second trimester that was maintained in the third following dietary health counseling performed by the midwife on an individual basis during the first trimester. These findings concur with those of O’Kane [53], who reported that women who received information on the importance of iodine during pregnancy increased their intake of this micronutrient. In contrast, Amiri [54] reported that knowledge of the importance of iodine and iodized salt intake was enhanced after an educational intervention in pregnant women, but there was no parallel improvement in their iodine status. Nonetheless, the authors advocated improving health literacy in pregnant women as an essential strategy.

Several studies, carried out in Spain, show that supplementation with KI during pregnancy makes it possible to achieve UIC ≥ 150 µg/L [40,41,42]. However, the study by Santiago et al. observed UIC values < 150 µg/L in pregnant women who consumed iodized salt for a year before gestation [40]. Menendez et al. [3] also obtained similar results among women who consumed > 2 glasses of milk. Likewise, our group obtained a median UIC > 150 in women who habitually consumed milk and iodized salt before pregnancy [4]. In the present study we observed how the consumption of iodized salt increases during pregnancy and that, like milk and iodized supplements, it maintains an independent effect to achieve UIC ≥ 150 µg/L. These data support the need to promote the consumption of dairy products and the use of iodized salt among women of childbearing age.

## 5. Limitations

This research faced the impossibility of carrying out the group of nutrition education interventions as planned in order to promote iodine consumption during pregnancy. As it has been mentioned, an individual nutrition education intervention was carried out in the midwife’s prenatal control surgery. However, the longitudinal design of the study allows us to attribute to it the changes observed in the iodine intake in the 2nd and 3rd trimester and its effects on the iodine status.

As for future research, the results of this study could be extended with a further study that would have the objective of establishing the impact of eating habits and iodine supplements on UIC amongst breastfeeding mothers postpartum.

## 6. Conclusions

In conclusion, the individual educational counseling on proper diet and nutrition carried out by the midwife during the routine first-trimester prenatal visit had a positive impact on the consumption of iodine-rich foods in pregnant women. Greater intake of milk and iodized salt, as well as KI supplementation, was associated with an increase in the UIC. As a result, the median UIC in the pregnant population in central Catalonia rose from the first to the second trimester of pregnancy and was sustained in the third. These data indicate adequate iodine nutrition in the pregnant population of this area, in accordance with the WHO, ICCIDD, and UNICEF recommendations.

Thus, a dietary educational intervention should be considered an essential component of pregnancy management to promote optimal iodine nutrition. The benefits obtained will contribute to decreasing fetal and maternal morbidity, as well as morbidity in the breast-fed infant. In addition, the knowledge gained can lead to better dietary habits in women beyond pregnancy.

## Figures and Tables

**Table 1 nutrients-12-02656-t001:** Characteristics and habits of the participants based on their responses to the dietary questionnaires.

	T1 (*n* = 945)	T2 (*n* = 752)	T3 (*n* = 705)	*p*
Age	30.6 (4.6)	30.4 (4.5)	30.4 (4.5)	0.726
Urban setting	363 (73.7%)	551 (73.3%)	523 (74.2%)	0.924
Educational level				
Illiterate	5 (0.5%)	5 (0.7%)	3 (0.4%)	0.999
Primary school education not completed	28 (3%)	18 (2.5%)	19 (2.8%)	
Primary school	232 (24.6%)	180 (24.8%)	165 (24.6%)	
High school	393 (41.6%)	297 (40.9%)	274 (40.9%)	
University degree	287 (30.4%)	226 (31.1%)	209 (31.2%)	
Place of origin				
Native of Spain	784 (83%)	600 (82.6%)	560 (83.6%)	0.999
South American	71 (7.5%)	53 (7.3%)	46 (6.9%)	
African	52 (5.5%)	43 (5.9%)	39 (5.8%)	
Other	38 (4%)	30 (4.1%)	25 (3.7%)	
Milk				
0 glasses	178 (18.8%)	103 (13.7%)	82 (11.6%)	<0.001
1–2 glasses	633 (67%)	525 (69.8%)	482 (68.4%)	
>2 glasses	134 (14.2%)	124 (16.5%)	141 (20%)	
Yogurt				
0 units	126 (13.3%)	80 (10.6%)	70 (9.9%)	0.070
1–2 units	192 (20.3%)	149 (19.8%)	124 (17.6%)	
>2 units	627 (66.3%)	523 (69.5%)	511 (72.5%)	
Cheese				
0 portions	131 (13.9%)	77 (10.2%)	80 (11.3%)	0.184
1–2 portions	214 (22.6%)	173 (23%)	153 (21.7%)	
>2 portions	600 (63.5%)	502 (66.8%)	472 (67%)	
Cooked vegetables				
0 servings	48 (5.1%)	24 (3.2%)	22 (3.1%)	0.030
1–2 servings	289 (30.6%)	212 (28.2%)	185 (26.2%)	
>2 servings	608 (64.3%)	516 (68.6%)	498 (70.6%)	
Fresh vegetables				
0 servings	51 (5.4%)	25 (3.3%)	21 (3%)	<0.001
1–2 servings	194 (20.5%)	109 (14.5%)	115 (16.3%)	
>2 servings	700 (74.1%)	618 (82.2%)	569 (80.7%)	
Fish				
0 servings	58 (6.1%)	29 (3.9%)	25 (3.5%)	<0.001
1–2 servings	535 (56.6%)	390 (51.9%)	335 (47.5%)	
>2 servings	352 (37.2%)	333 (44.3%)	345 (48.9%)	
Canned fish				
0 servings	142 (15%)	77 (10.2%)	72 (10.2%)	<0.001
1–2 servings	495 (52.4%)	351 (46.7%)	308 (43.7%)	
>2 servings	308 (32.6%)	324 (43.1%)	325 (46.1%)	
Meat				
0 servings	36 (3.8%)	11 (1.5%)	12 (1.7%)	0.016
1–2 servings	123 (13%)	97 (12.9%)	89 (12.6%)	
>2 servings	786 (83.2%)	644 (85.6%)	604 (85.7%)	
Cold cooked meat				
0 servings	133 (14.1%)	127 (16.9%)	115 (16.3%)	0.170
1–2 servings	261 (27.6%)	212 (28.2%)	218 (30.9%)	
>2 servings	551 (58.3%)	413 (54.9%)	372 (52.8%)	
Eggs				
0 servings	52 (5.5%)	18 (2.4%)	22 (3.1%)	<0.001
1–2 servings	526 (55.7%)	399 (53.1%)	356 (50.5%)	
>2 servings	367 (38.8%)	335 (44.5%)	327 (46.4%)	
Fruit				
0 servings	38 (4%)	17 (2.3%)	11 (1.6%)	<0.001
1–2 servings	89 (9.4%)	46 (6.1%)	35 (5%)	
>2 servings	818 (86.6%)	689 (91.6%)	659 (93.5%)	
Dried fruit				
0 servings	366 (38.7%)	286 (38%)	284 (40.3%)	0.918
1–2 servings	351 (37.1%)	285 (37.9%)	260 (36.9%)	
>2 servings	228 (24.1%)	181 (24.1%)	161 (22.8%)	
Iodized salt	337 (35.7%)	622 (85.7%)	598 (89.3%)	<0.001
Iodine supplementation	442 (46.8%)	645 (88.8%)	587 (87.6%)	<0.001
Tobacco consumption				0.042
Non smoker	626 (66.2%)	542 (67.8%)	545 (68%)	
Ex-smoker	98 (10.4%)	105 (13.1%)	106 (13.2%)	
Smoker	221 (23.4%)	153 (19.1%)	150 (18.7%)	
UIC, µg/L (*)	172 (104–289)	210 (130–319)	202.5 (131–331)	<0.001
UIC ≥ 150 µg/L	556 (57.3%)	514 (68.5%)	479 (68%)	<0.001

(*) Median and first-third quartiles (Q1–Q3). UIC, urinary iodine concentration; T1, T2, and T3, first, second, and third trimester.

**Table 2 nutrients-12-02656-t002:** Percentage of pregnant women with UIC ≥ 150 µg/dL related to dietary intake, iodized salt intake, iodine supplementation, and smoking.

	T1	*p*	T2	*p*	T3	*p*
	*n* = 945		*n* = 752		*n* = 705	
Milk		0.016		0.129		0.543
0 glasses	85/178 (47.8%)		63/103 (61.2%)		51/81 (63%)	
1–2 glasses	372/633 (58.8%)		363/523 (69.4%)		332/482 (68.9%)	
>2 glasses	81/134 (60.4%)		88/124 (71%)		96/141 (68.1%)	
Yogurt		0.372		0.468		0.416
0 units	72/126 (57.1%)		57/80 (71.3%)		46/70 (65.7%)	
1–2 units	99/192 (51.6%)		104/149 (69.8%)		81/124 (65.3%)	
>2 units	367/627 (58.5%)		353/521 (67.8%)		352/510 (69%)	
Cheese		0.104		0.689		0.903
0 portions	69/131 (52.7%)		51/77 (66.2%)		53/80 (66.3%)	
1–2 portions	115/214 (53.7%)		118/172 (68.6%)		108/153 (70.6%)	
>2 portions	354/600 (59%)		345/501 (68.9%)		318/471 (67.5%)	
Cooked vegetables		0.561		0.715		0.046
0 servings	26/48 (54.2%)		12/24 (50%)		11/22 (50%)	
1–2 servings	162/289 (56.1%)		151/211 (71.6%)		120/184 (65.2%)	
>2 servings	350/608 (57.6%)		351/515 (68.2%)		348/498 (69.9%)	
Fresh vegetables		0.024		0.630		0.140
0 servings	21/51 (41.2%)		15/25 (60%)		9/21 (42.9%)	
1–2 servings	107/194 (55.2%)		76/109 (69.7%)		80/115 (69.6%)	
>2 servings	410/700 (58.6%)		423/616 (68.7%)		390/568 (68.7%)	
Fish		0.277		0.927		0.413
0 servings	31/58 (53.4%)		15/28 (53.6%)		16/25 (64%)	
1–2 servings	299/535 (55.9%)		275/390 (70.5%)		235/334 (70.4%)	
>2 servings	208/352 (59.1%)		224/332 (67.5%)		228/345 (66.1%)	
Canned fish		0.184		0.515		0.578
0 servings	72/142 (50.7%)		50/77 (64.9%)		50/72 (69.4%)	
1–2 servings	286/495 (57.8%)		240/350 (68.6%)		203/308 (65.9%)	
>2 servings	180/308 (58.4%)		224/323 (69.3%)		226/324 (69.8%)	
Meat		0.590		0.147		0.678
0 servings	18/36 (50%)		6/11 (54.5%)		8/12 (66.7%)	
1–2 servings	71/123 (57.7%)		77/97 (79.4%)		63/89 (70.8%)	
>2 servings	449/786 (57.1%)		431/642 (67.1%)		408/603 (67.7%)	
Cold cooked meat		0.529		0.013		0.623
0 servings	74/133 (55.6%)		97/127 (76.4%)		76/114 (66.7%)	
1–2 servings	159/261 (60.9%)		148/210 (70.5%)		156/218 (71.6%)	
>2 servings	305/551 (55.4%)		269/413 (65.1%)		247/372 (66.4%)	
Eggs		0.383		0.799		0.718
0 servings	32/52 (61.5%)		11/18 (61.1%)		14/22 (63.6%)	
1–2 servings	287/526 (54.6%)		273/397 (68.8%)		241/355 (67.9%)	
>2 servings	219/367 (59.7%)		230/335 (68.7%)		224/327 (68.5%)	
Fruit		0.027		0.862		0.195
0 servings	17/38 (44.7%)		12/17 (70.6%)		9/11 (81.8%)	
1–2 servings	44/89 (49.4%)		30/46 (65.2%)		26/35 (74.3%)	
>2 servings	477/818 (58.3%)		472/687 (68.7%)		444/658 (67.5%)	
Dried fruit		0.639		0.078		0.653
0 servings	199/366 (54.4%)		198/284 (69.7%)		189/284 (66.5%)	
1–2 servings	213/351 (60.7%)		206/285 (72.3%)		189/259 (73%)	
>2 servings	126/228 (55.3%)		110/181 (60.8%)		101/161 (62.7%)	
Iodized salt		0.001		0.030		0.004
No	321/608 (52.8%)		65/109 (59.6%)		42/78 (53.8%)	
Yes	217/337 (64.4%)		449/641 (70%)		437/626 (69.8%)	
Iodine supplementation		<0.001		0.002		<0.001
No	234/503 (46.5%)		45/84 (53.6%)		44/87 (50.6%)	
Yes	304/442 (68.8%)		469/666 (70.4%)		435/617 (70.5%)	
Tobacco use		0.178		0.247		0.908
Non-smoker	374/633 (59.1%)		349/520 (67.1%)		331/485 (68.2%)	
Smoker	54/98 (55.1%)		54/75 (72%)		53/75 (70.7%)	
Ex-smoker	123/232 (53%)		110/154 (71.4%)		95/144 (66%	

**Table 3 nutrients-12-02656-t003:** Individual effects of each variable included in multilevel mixed-effects logistic regression models adjusted for paired measures in the 3 trimesters to assess the effect of iodine-rich food intake or supplementation on UIC values < 150 μg/L.

UIC	β Coefficient	OR (95% CI)	*p*	1/OR (95% CI)
Milk				
1–2 glasses	−0.464	0.63 (0.48–0.82)	0.001	1.59 (2.08–1.22)
>2 glasses	−0.531	0.59 (0.42–0.83)	0.002	1.70 (2.40–1.21)
Yogurt				
1–2 units	0.134	1.14 (0.80–1.63)	0.461	0.87 (1.25–0.61)
>2 units	−0.067	0.94 (0.69–1.27)	0.670	1.07 (1.45–0.79)
Cheese				
1–2 portions	−0.164	0.85 (0.60–1.19)	0.345	1.18 (1.65–0.84)
>2 portions	−0.238	0.79 (0.58–1.07)	0.122	1.27 (1.72–0.94)
Cooked vegetables				
1–2 portions	−0.510	0.60 (0.36–0.99)	0.046	1.67 (2.75–1.01)
>2 portions	−0.581	0.56 (0.34–0.91)	0.019	1.79 (2.90–1.10)
Fresh vegetables				
1–2 portions	−0.674	0.51 (0.31–0.85)	0.009	1.96 (3.26–1.18)
>2 portions	−0.812	0.44 (0.28–0.71)	0.001	2.25 (3.61–1.41)
Fish				
1–2 portions	−0.363	0.70 (0.44–1.09)	0.113	1.44 (2.25–0.92)
>2 portions	−0.358	0.70 (0.44–1.10)	0.124	1.43 (2.26–0.91)
Canned fish				
1–2 portions	−0.193	0.82 (0.61–1.12)	0.212	1.21 (1.64–0.90)
>2 portions	−0.372	0.69 (0.50–0.94)	0.021	1.45 (1.99–1.06)
Meet				
1–2 portions	−0.636	0.53 (0.27–1.02)	0.057	1.89 (3.64–0.98)
>2 portions	−0.413	0.66 (0.36–1.21)	0.183	1.51 (2.78–0.82)
Cold cooked meat				
1–2 portions	−0.100	0.90 (0.66–1.23)	0.525	1.11 (1.51–0.81)
>2 portions	0.188	1.21 (0.91–1.60)	0.192	0.83 (1.10–0.62)
Eggs				
1–2 portions	0	1 (0.61–1.65)	1.000	1 (1.65–0.61)
>2 portions	−0.156	0.86 (0.52–1.42)	0.545	1.17 (1.94–0.71)
Fruit				
1–2 portions	−0.104	0.90 (0.47–1.74)	0.757	1.11 (2.14–0.57)
>2 portions	−0.348	0.71 (0.40–1.25)	0.232	1.42 (2.51–0.80)
Dried fruit				
1–2 portions	−0.221	0.80 (0.64–1.00)	0.050	1.25 (1.56–1.00)
>2 portions	0.147	1.16 (0.90–1.49)	0.248	0.86 (1.11–0.67)
Iodized salt				
yes	−0.726	0.48 (0.40–0.59)	0	2.07 (2.53–1.69)
Iodine supplementation				
yes	−1.035	0.36 (0.29–0.44)	0	2.82 (3.48–2.28)
Tobacco use				
Ex-smoker	0.012	1.01 (0.72–1.42)	0.944	0.99 (1.38–0.71)
Smoker	0.170	1.19 (0.93–1.51)	0.173	0.84 (1.08–0.66)

**Table 4 nutrients-12-02656-t004:** Results of multilevel mixed-effects logistic regression models adjusted for paired measures in the 3 trimesters to assess the effect of iodine-rich food intake or supplementation on UIC values <150 μg/L.

A. Multilevel Mixed-Effects				
**Variables**	**Coefficient**	**OR (95% CI)**	***p***	**1/OR (95% CI)**
Iodine supplementation	−0.872	0.42 (0.34–0.52)	0	2.39 (1.92–2.98)
Iodized salt	−0.403	0.67 (0.54–0.83)	0	1.50 (1.21–1.85)
Milk				
1–2 glasses	−0.383	0.68 (0.52–0.90)	0.006	1.47 (1.12–1.93)
>2 glasses	−0.453	0.64 (0.45–0.90)	0.011	1.57 (1.11–2.23)
Constant	0.575			
**B. Multilevel with Imputation**				
**Variables**	**Coefficient**	**OR (95% CI)**	***p***	**1/OR (95% CI)**
Iodine supplementation	−0.845	0.43 (0.35–0.53)	0	2.33 (1.89–2.87)
Iodized salt	−0.383	0.68 (0.55–0.84)	0	1.47 (1.19–1.81)
Milk				
1–2 glasses	−0.382	0.68 (0.53–0.88)	0.003	1.46 (1.13–1.89)
>2 glasses	−0.456	0.63 (0.45–0.89)	0.008	1.58 (1.13–2.21)
Constant	0.567			

**Table 5 nutrients-12-02656-t005:** Summary of the missing data imputation.

Variable	Complete	Imputed	Total
Iodine	2338 (82.5%)	497 (17.5%)	2835
Milk	2341 (82.6%)	494 (17.4%)	2835
Fresh vegetables	2341 (82.6%)	494 (17.4%)	2835
Cooked vegetables	2341 (82.6%)	494 (17.4%)	2835
Canned fish	2341 (82.6%)	494 (17.4%)	2835
Tobacco use	2546 (89.8%)	289 (10.2%)	2835
Iodine supplementation	2341 (82.6%)	494 (17.4%)	2835
Iodized salt	2341 (82.6%)	494 (17.4%)	2835

## Data Availability

The data supporting the conclusions of this study are available upon reasonable request and under the supervision of IDIAP Jordi Gol.

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
