# Peer review of "Impact of Dietary Habit, Iodine Supplementation and Smoking Habit on Urinary Iodine Concentration During Pregnancy in a Catalonia Population"

_nutrients, 2020, doi:10.3390/nu12092656_

Round 1

Reviewer 1 Report

The Authors conducted an interesting and well-constructed study on the various factors affecting iodine concentration during pregnancy, in Catalonia population.

Few observations could be made, to improve the manuscript.

 Abstract:

  • Clarify what KI stands for or use the non-abbreviated form
  • Since in the multivariate analysis fresh vegetables and fruit did not have an impact o iodine status, it should be appropriate to avoid this also from the abstract, in order to not give misleading messages

Introduction

  • “To cover iodine needs during pregnancy, the diet should contain 3 servings of milk and dairy products and 3 g of iodized salt per day”. What is the reference for this?
  • There is still discussion whether iodine supplementation is warranted during pregnancy in developed areas where iodized salt is available. American Thyroid Association recommends iodine supplementation during pregnancy (150 ug/day), while WHO states that the supplementation can be avoid in areas in which iodized salt is spread to at least 90% of households and median UIC is > 100 ug/L (iodine sufficient areas) (WHO Secretariat on behalf of the participants to the Consultation, Public Health Nutrition 2007). In what of these WHO categories does Spain fit into?

Results

  • The Authors state that 97.4% answered the dietary questionnaire at the first-trimester visit, 77.5% al the second-trimester visit and 72.7% at the third-trimester visit, but what about the urine collection? Was the number of available first-morning urine samples the same as the number of questionnaires? This is important to know to understand possible biases between data collected with the questionnaire and UIC values.
  • According to the discussion about the necessity or not of an iodine supplement during pregnany and the effective contribution of the diet, it would be very very interesting to analyse the effect of dietary education independently from iodine supplement. Is this data available? This data would be useful to really understand if the dietary education is effective and if iodine supplement can be avoided.
  • Why is table 5 cited in the manuscript before table 4?
  • Table 2: In the column of quantities (i.e. milk 0 galsses; 1-2 glasses and >2 glasses) I would suggest to add the total number of samples included (i.e.: milk 0 glasses: 178 and so on). This is useful to understand more rapidly what the various numbers and percentages indicated in the tables refer to.

Discussion

  • UNICEF anagram significance was already explained in the Introduction section
  • Various “μ” were delated in the text, maybe because of informatic troubles
  • Together with Amiri et al study, also another randomized controlled data (Censi et al, Nutrients 2019) found parallel results. It seems that dietary education of pregnant women can improve iodine status, but not reaching a satisfying UIC (≥150 ug/L). Thus, the iodine supplement seems to be necessary to reach this goal in pregnant women. What about Catalonia area? In the 3 studies carried out in Spain in 2013 and 2009 and also cited by the Authors (40-42), KI supplementation was necessary to increase UIC. If this data is available (see Results point 2) it would be very interesting to comment.

Limitations:

  • An extra “I” in the title

Author Response

RESPONSE TO REVIEWERS

REVIEWER 1

The Authors conducted an interesting and well-constructed study on the various factors affecting iodine concentration during pregnancy, in Catalonia population.

Few observations could be made, to improve the manuscript.

 Abstract

  1. Clarify what KI stands for or use the non-abbreviated form

In the abstract we have changed "KI" to “potassium iodide” (page 1)

  1. Since in the multivariate analysis fresh vegetables and fruit did not have an impact on iodine status, it should be appropriate to avoid this also from the abstract, in order to not give misleading messages

The reviewer is right, it can create confusion. We have removed these foods from the abstract. (page 1)

Introduction

  1. To cover iodine needs during pregnancy, the diet should contain 3 servings of milk and dairy products and 3 g of iodized salt per day”. What is the reference for this?

 We have included in the text the references that support this statement (3, 4, 10)(èpage 2)

  1. There is still discussion whether iodine supplementation is warranted during pregnancy in developed areas where iodized salt is available. American Thyroid Association recommends iodine supplementation during pregnancy (150 ug/day), while WHO states that the supplementation can be avoid in areas in which iodized salt is spread to at least 90% of households and median UIC is > 100 ug/L (iodine sufficient areas) (WHO Secretariat on behalf of the participants to the Consultation, Public Health Nutrition 2007). In what of these WHO categories does Spain fit into?

The situation in Spain is dual: the consumption of iodized salt in Spain does not reach 90%, the average consumption is 69%. However, the median UIC is > 100 µg/L (117 µg /L in the adult population and 173 µg/L in the child population). So if iodine intake is not guaranteed through food in pregnant women, supplementation with KI should be recommended.

Results

  1. The Authors state that 97.4% answered the dietary questionnaire at the first-trimester visit, 77.5% al the second-trimester visit and 72.7% at the third-trimester visit, but what about the urine collection? Was the number of available first-morning urine samples the same as the number of questionnaires? This is important to know to understand possible biases between data collected with the questionnaire and UIC values.

The reviewer is right. It is important information to consider. In each trimester we had more iodine samples collected than questionnaires conducted. So the limiting factor is the dietary questionnaire. In the first paragraph of the results, we indicated that there were no sociodemographic differences between participants with full follow-up and those without. But as you indicate, it is important to know that there were no differences in terms of UIC between the two groups. For this reason, we have added this information in the same paragraph (“nor for UIC”)(page 4)

  1. According to the discussion about the necessity or not of an iodine supplement during pregnancy and the effective contribution of the diet, it would be very interesting to analyse the effect of dietary education independently from iodine supplement. Is this data available? This data would be useful to really understand if the dietary education is effective and if iodine supplement can be avoided.

Unfortunately, we do not have this information. In the Materials and Methods section (Design), we discussed this issue. “Initially, the study was contemplated as part of a clinical trial project to evaluate the efficacy of an educational intervention imparted in groups to promote iodine intake and optimal urinary iodine concentrations (≥ 150 μg/L) in pregnant women. The women were enrolled, but the intervention by groups was found to be impracticable in our setting because of difficulties to ensure proper development of the educational program. Hence, the design was converted to an observational study with a single-arm. However, the results of our study reflect a positive impact of individual health education on the consumption of iodized salt, milk, KI supplementation, and tobacco consumption in the pregnant women studied. The most relevant effect is the increase in UIC from the first to the second quarter and its maintenance in the third.

  1. Why is table 5 cited in the manuscript before table 4?

The reviewer is right. We have ordered the numeration (ètables 4 and 5).

  1. Table 2: In the column of quantities (i.e. milk 0 glasses; 1-2 glasses and >2 glasses) I would suggest to add the total number of samples included (i.e.: milk 0 glasses: 178 and so on). This is useful to understand more rapidly what the various numbers and percentages indicated in the tables refer to.

We have added the denominators related to the calculation of each percentage. (Table 2)

Discussion

  1. UNICEF anagram significance was already explained in the Introduction section

We have removed the repeated explanation from the acronym

  1. Various “μ” were delated in the text, maybe because of informatic troubles

We have reviewed and corrected the error

  1. Together with Amiri et al study, also another randomized controlled data (Censi et al, Nutrients 2019) found parallel results. It seems that dietary education of pregnant women can improve iodine status, but not reaching a satisfying UIC (≥150 ug/L). Thus, the iodine supplement seems to be necessary to reach this goal in pregnant women. What about Catalonia area? In the 3 studies carried out in Spain in 2013 and 2009 and also cited by the Authors (40-42), KI supplementation was necessary to increase UIC. If this data is available (see Results point 2) it would be very interesting to comment.

Following the reviewer's suggestion, we have extended the discussion to comment on this aspect:

Several studies, carried out in Spain, show that supplementation with KI during pregnancy makes it possible to achieve UIC ≥ 150 µg/L [40-42]. However, the study by Santiago et al observed UIC values < 150 µg/L in pregnant women who consumed iodized salt for a year before gestation [40]. Menendez et al [3] also obtained similar results among women who consumed >2 glasses of milk. Likewise, our group obtained a median UIC >150 in women who habitually consumed milk and iodized salt before pregnancy [4]. In the present study we observed how the consumption of iodized salt increases during pregnancy and that, like milk and iodized supplements, it maintains an independent effect to achieve UIC ≥150 µg/L. These data support the need to promote the consumption of dairy products and the use of iodized salt among women of childbearing age.”

Limitations

  1. An extra “I” in the title

We have reviewed and corrected the error

Reviewer 2 Report

The paper by Torres and co-Authors is sound and quite well written, unfortunately it refers to an observational study conducted between years 2009-2011 so the nutritional status in pregnancy in their area of Catalonia is not contemporary but refers to around ten years ago.

I have just minor remarks:

Table 1 contains a lot of data but and it is quite hypertrophic, it is more appropriate that the Authors present only dietary habits changing along different trimesters moreover significance levels (p values) do not figure in the table for iodized salt, iodine supplementation, tobacco consumption.

What is the concentration of iodine in supplements used in pregnancy?

Authors recorded women of differential ethnicities in their series; is ethnicity a factor influencing iodine Status? A study carried out in the same year by quite similar modalities (Mian et all Clinical Endocrinology 2009) showed that foreign women had a poorer iodine status because of the lower use of iodized salt and supplements and differential type of food. Please give a comment about this in Discussion.

Could the Authors give some data by separating different groups of women regarding differential use of iodized salt (yes or not) and iodine supplement (yes or not) according to the differential consumption of milk (1-2 glasses versus >2 glasses) comparing differential median values by Kruskal-Wallis?

Author Response

RESPONSE TO REVIEWERS

REVIEWER 2

Comments and Suggestions for Authors

The paper by Torres and co-Authors is sound and quite well written, unfortunately it refers to an observational study conducted between years 2009-2011 so the nutritional status in pregnancy in their area of Catalonia is not contemporary but refers to around ten years ago.

I have just minor remarks:

  1. Table 1 contains a lot of data but and it is quite hypertrophic, it is more appropriate that the Authors present only dietary habits changing along different trimesters moreover significance levels (p values) do not figure in the table for iodized salt, iodine supplementation, tobacco consumption.

The reviewer is correct, it is a hypertrophic table, but its structure is also related to the information in table 2. Despite this, we consider it interesting to show all the information, since all educational intervention includes all the variables shown.

For that reason, if the reviewer doesn't see it as a major inconvenience, we would like to keep it as is.

We have added the significance values that did not appear in Table 1. We have detected a problem with the tobacco consumption data, which we have reviewed and updated (table 1)

  1. What is the concentration of iodine in supplements used in pregnancy?

In our territory, doses of 100 to 300 µgr are marketed. The recommended dose for women in our public system is 200-250 µgr. In the introduction we provide this information and the reference to the bibliographic reference 10 (page 2)

  1. Authors recorded women of differential ethnicities in their series; is ethnicity a factor influencing iodine Status? A study carried out in the same year by quite similar modalities (Mian et all Clinical Endocrinology 2009) showed that foreign women had a poorer iodine status because of the lower use of iodized salt and supplements and differential type of food. Please give a comment about this in Discussion.

In our study, ethnicity was never a statistically significant factor at any time. It has not been included in the results as we had a small sample of immigrants from many different origins. Although the categories shown in table 1 seem an interesting sample, in reality each one of them groups different ethnic groups (for example, the category “African” includes both Maghreb and sub-Saharan Africans)

Thus, the sample of migrant women is of very diverse origins and too small in size

  1. Could the Authors give some data by separating different groups of women regarding differential use of iodized salt (yes or not) and iodine supplement (yes or not) according to the differential consumption of milk (1-2 glasses versus >2 glasses) comparing differential median values by Kruskal-Wallis?

We are not sure we understand your suggestion. The Kruskal-Wallis test allows bivariate analysis, in this case to analyze the relationship of the outcome (UIC) with each of the independent factors (iodine salt, iodine supplement and consumption of milk). But for each factor analyzed, we do not know its individual effect once adjusted for the effect of the other factors. That is why it is especially important to observe the adjusted effect in a multivariate analysis like the one we have carried out.

We are in front of a repeated measures design over 3 trimesters, so we believe that bivariate analysis is not adequate.

However, if what the reviewer is suggesting is to show a table of a summary with the UIC medians for each quarter, for each factor, we could build a new table with this information. If the reviewer finds it interesting, a solution would be to consider a table attached to the article.

If we have not understood your proposal correctly, we will be grateful if you can provide us with more information

Round 2

Reviewer 1 Report

Authors have satisfyingly addressed the questions.